# Social Support and Resilience as Predictors of Prosocial Behaviors before and during COVID-19

**DOI:** 10.3390/healthcare10091669

**Published:** 2022-09-01

**Authors:** Javier Esparza-Reig, Manuel Martí-Vilar, Francisco González-Sala, César Merino-Soto, Filiberto Toledano-Toledano

**Affiliations:** 1Departamento de Psicología, Universidad Europea de Valencia, Passeig de l’Albereda 7, 46010 Valencia, Spain; 2Departamento de Psicología Básica, Universitat de València, Avgda. Blasco Ibañez 21, 46010 Valencia, Spain; 3Departamento de Psicología Evolutiva y de la Educación, Universitat de València, Avgda. Blasco Ibañez 21, 46010 Valencia, Spain; 4Instituto de Investigación de Psicología, Universidad de San Martín de Porres (Perú), Av. Tomás Marsano 232, Lima 34, Peru; 5Unidad de Investigación en Medicina Basada en Evidencias, Hospital Infantil de México Federico Gómez, National Institute of Health, Dr. Márquez 162, Doctores, Cuauhtémoc, Mexico City 06720, Mexico; 6Unidad de Investigación Sociomédica, Instituto Nacional de Rehabilitación Luis Guillermo Ibarra Ibarra, Calzada México-Xochimilco 289, Arenal de Guadalupe, Tlalpan, Mexico City 14389, Mexico

**Keywords:** prosocial behavior, Bayesian statistics, resilience, social support, COVID-19

## Abstract

The objective of this research was to analyze the relationship between social support and resilience with prosocial behavior before and during the confinement caused by COVID-19. **Materials and Methods**: The participants were divided into a confined group (228 women and 84 men) and an unconfined group (153 women and 105 men), all of whom were university students. Instruments were applied to measure the variables proposed. **Results:** Social support predicted 24.4% of the variance in prosocial behavior among women and 12% among men in the confined group; no evidence of this relationship was found in the unconfined groups. Resilience predicted 7% of the variance in prosocial behavior among confined women, 8.4% among confined men, 8.8% among unconfined women, and 5.1% in unconfined men. **Discussion and Conclusion:** The results show the importance of social support and resilience in prosocial behaviors, which are key elements for the proper functioning of society, especially in the face of a crisis such as COVID-19.

## 1. Introduction

Prosocial behaviors aim to benefit or help other people or groups voluntarily, and without seeking anything in return [1]. These behaviors are fundamental to making a society function properly [2], and in addition to helping others directly, they reduce aggression and other antisocial behaviors in society [3], and even act as a protective factor against the development of substance addiction problems [4] and gambling [5,6].

Prosocial behaviors facilitate the perception of support among individuals. Various studies have found that feeling supported by individuals who are part of ones’ social network is related to the prosocial behaviors that people demonstrate toward others; therefore, perceived social support is a predictor of prosocial behaviors [5,7]. Conversely, feeling that one does not have support in one’s environment seems to be related to problems that trigger antisocial behaviors that are harmful to the rest of society [8].

Another aspect involved in prosocial behaviors is the ability to cope with problems and adversities that may arise throughout life, i.e., resilience. Resilience has been shown to be directly related to people’s prosocial behaviors [9,10], and encourages individuals to manifest more prosocial values and increase prosocial behaviors instead of focusing on themselves [5,11].

Although prosocial behaviors are always necessary for the proper functioning of a society, their importance increases during difficult periods faced by societies, such as natural disasters [12]. Thus, favoring and enhancing collective values instead of individualistic ones, not focusing on oneself and thinking of others, is a key aspect of the ability to overcome adversity as a society [13].

Currently, the global pandemic caused by COVID-19 [14] is an unprecedented crisis that all societies must face to try to minimize the consequences. These consequences are not limited to the most obvious health consequences evidenced by the growing number of infected people and deaths, but they also causing mental health issues due to the stressors that individuals are enduring; further, this crisis and the measures adopted by governments to curb the disease are having strong economic and social repercussions [15,16].

The effect of the pandemic and quarantine measures on prosocial behaviors has not yet been investigated in depth, but some authors [17] point out that quarantine has decreased the occasions for prosocial behaviors in adolescents, but not their intention to engage in them.

The links between social support and health during quarantine have been studied to a great extent. Existing research indicates that during the period of social isolation enacted in multiple countries as a precautionary measure, social support acted as a protective factor against problematic alcohol consumption [18]. Lower perceived social support is associated with mental health problems such as depression and anxiety [19], and higher rates of social support are associated with better stress coping strategies, which is particularly important during isolation and quarantine when considering all the stressors generated by this situation worldwide [20].

Finally, it is very important to consider the challenge that this pandemic poses for all humanity, and the need for all people to develop resilience so they can face these novel difficulties as individuals and as a society [21,22]. Various studies have shown that resilience is directly related to the nonappearance of anxiety in health care personnel during this crisis [23], and is a protective factor against the development of anxiety and depression during quarantine [24].

The present research aims to analyze the relationship between perceived social support and resilience with prosocial behaviors in men and women without confinement, and during the quarantine decreed by Spain in response to the COVID-19 outbreak there.

Based on this objective, the first hypothesis is that regardless of the sex of the participants, perceived social support will be directly related to prosocial behaviors in both the confined and non-confined groups. Furthermore, social support will act as a protective factor favoring prosocial behaviors.

The second hypothesis is that in the case of resilience, there will also be the same direct correlation with prosocial behaviors in men and women, regardless of the confinement status. Additionally, in this case, people’s resilience will be a protective factor that will enhance their prosocial behaviors.

## 2. Materials and Methods

### 2.1. Participants

Sampling for this study was incidental, and missing values and outliers were eliminated. After sample purification by eliminating missing values and outliers, the sample without home confinement (prior to the COVID-19 outbreak) consisted of 153 women with a mean age of 20.76 (*SD* = 2.05) years and 105 men with a mean age of 21.23 (*SD* = 2.37) years. The sample that was subjected to in-home confinement consisted of 228 women with a mean age of 22.26 (*SD* = 2.26) years and 84 men with a mean age of 23.14 (*SD* = 2.07) years; these participants had been confined for 22 to 53 days, with a mean of 27.64 (*SD* = 6.21) days for women and 23 to 54 days, with a mean of 26.89 (*SD* = 6.23) days for men. Participants were university students residing in Spain, aged 18 to 26 years.

### 2.2. Instruments

First, the social support questionnaire (MOS; [25]) was applied in its validated Spanish version [26]. Respondents are asked for the number of family members and close friends, to quantify their social networks. They then respond to the 19-item questionnaire that measures the perception of social support on a 5-point Likert-type scale. For this research, only the total scores on this instrument were analyzed. In the unconfined sample, the Cronbach’s alpha was 0.94 for both males and females, while in the confined sample, it was 0.96 for males and 0.95 for females.

Next, the Brief Resilient Coping Scale (BRCS; [27]) was applied in its validated Spanish version [28]. This scale measures the resilience capacity of respondents by means of 4 Likert-type items with 5 response choices. In this case, the Cronbach’s alpha values for the unconfined group were 0.59 for men and 0.61 for women, while for the confined group, they were 0.68 among men and 0.56 among women.

Finally, the Prosociality Scale [29] adapted for the Spanish population [30] was used to evaluate participants’ prosocial behaviors. This scale is made up of 16 items measured on a 5-point Likert-type scale, in which higher scores show higher rates of prosocial behavior. In the nonconfined group, Cronbach’s alpha values were 0.89 for men and 0.88 for women, while in the confined group, they were 0.92 in men and 0.84 in women.

### 2.3. Procedures

This research was approved by the Ethics Commission on Experimental Research at the University of Valencia (procedure number 1040164). The first sample was collected prior to the crisis caused by COVID-19, from May to December 2019. All participants were university students residing in Spain who completed the questionnaire in person on paper, always in the presence of one of the researchers. All participants signed an informed consent form that described the investigation and guaranteed the anonymity of the data. The questionnaire took approximately 50 min to complete. Participants did not receive any incentive.

The second sample was collected from March to May 2020, a period in which all the participants were confined to their homes, as decreed by the government. In this case, the questionnaire had to be completed online because the exceptional situation made it impossible to do otherwise. Before answering the questionnaire, the participants had to read and accept an informed consent form, which presented the conditions of the research and guaranteed the anonymity of the data.

### 2.4. Statistical Analysis

First, the sampling distributions and response frequencies of confined and unconfined men and women were analyzed.

Pearson correlation analyses and simple linear regressions were then performed from the Bayesian statistics approach following the recommendations of Depaoli and Van de Schoot [31], and using uninformed a priori distributions with the Jeffreys Zellner Siow (JZS) method for greater precision [32].

The use of Bayesian statistics instead of the classical Frequentist approach has as its main advantages the incorporation of prior knowledge (a priori information) obtained in previous research, the ability to work with small samples, since it does not depend on sample size as in the Frequentist approach, and no testing of strong parametric assumptions; since the interval estimates are more precise, it offers probability or credibility intervals instead of confidence intervals and calculates the probability of hypotheses in hypothesis testing [33].

The interpretation of the Bayes factor (BF) values was performed according to the criteria used by different authors [32,34]. All analyses were performed using SPSS v.25, (IBM Inc., Chicago, IL, USA).

## 3. Results

The mean scores obtained by each group of participants for the variables of social support, resilience, and prosocial behavior are shown in Table 1.

### 3.1. Bayesian Correlation Analysis

The analysis of correlations of prosocial behavior with social support and resilience is presented in Table 2. In the case of the relationship between prosocial behavior and social support, the results showed extreme evidence in favor of the alternative hypothesis (H1) versus the null hypothesis (H0). That is, there was extreme evidence of the existence of a correlation between both variables in the group of confined women (FB01 = 0.00), and there was strong evidence that this relationship existed in the group of confined men (FB01 = 0.04). For both the nonconfined group of women (FB01 = 2.20) and men (FB01 = 1.82), anecdotal evidence was found in favor of H0, i.e., in favor of the nonexistence of a correlation between prosocial behavior and social support.

In the case of the analysis of correlations between prosocial behavior and resilience, extreme evidence of the existence of this correlation was found in the groups of nonconfined (FB01 = 0.01) and confined (FB01 = 0.00) women. Moderate evidence of the existence of this relationship was found in the group of confined men (FB01 = 0.20) and anecdotal evidence of the relationship in the case of nonconfined men (FB01 = 0.56). Table 2 also presents the a posteriori values of mode, mean, and variance, and the 95% confidence interval (CI).

### 3.2. Simple Bayesian Linear Regression Analysis

First, regarding the regression analysis of the relationship between prosocial behavior as a dependent variable (DV) and social support as an independent variable (IV), the results showed extreme evidence in favor of H1 (the existence of this relationship) in the group of confined women (FB10 = 3.7712), and strong evidence for the existence of this relationship in the group of confined men (FB10 = 25.57). Anecdotal evidence in favor of the nonexistence of this relationship was found in the group of unconfined women (FB10 = 0.45) and in the group of unconfined men (FB10 = 0.55).

In the groups of unconfined participants, the model analyzed was significant in both women (F (1) = 3.99, *p* < 0.05) and men (F (1) = 4.04, *p* < 0.05), explaining 1.9% and 2.8% of the variance, respectively. In the case of confined females, the model was significant (F (1) = 74.08, *p* < 0.001) and explained 24.4% of the variance in prosocial behaviors, while in the group of confined males, it was also significant (F (1) = 12.36, *p* < 0.01) and explained 12% of the variance.

Second, regression analysis of the relationship between prosocial behavior as DV and resilience as IV reflected extreme evidence in favor of the existence of the relationship in the unconfined (FB10 = 103.07) and confined (FB10 = 317.75) female groups. For the male groups, moderate evidence in favor of H1 was found in the confined males (FB10 = 5.02), and anecdotal evidence in favor of H1 was found in the unconfined males (FB10 = 1.79).

Additionally, in this case, the model proposed was significant in all groups and explained 8.8% of the variance in prosocial behaviors in the group of unconfined women (F (1) = 15.61 *p* < 0.01), 5.1% in the unconfined men (F (1) = 6.55, *p* < 0.05), 7% in the confined female group (F (1) = 18.18, *p* < 0.01), and 8.4% in the confined male group (F (1) = 8.65, *p* < 0.01). Finally, Table 3 presents the a posteriori values of the mode, mean, and variance, and the 95% CIs.

## 4. Discussion

The main objective of this research was to analyze the relationship between perceived social support and resilience with prosocial behaviors in men and women, with and without home confinement decreed by the Spanish government as a precaution against COVID-19. The aim of this study was to deepen our knowledge of these relationships by conducting research under different conditions.

Based on this objective, the first hypothesis was that prosocial behavior would be directly related to perceived social support. This relationship would occur in men and women, both with and without home confinement. Additionally, it was proposed that perceived social support would be a protective factor that would favor prosocial behaviors. The results obtained in the confined groups are congruent with this hypothesis, since they reflect extreme evidence of this direct relationship in women, and strong evidence in men. In both groups, extreme evidence was also found for the female group and strong evidence for the male group that perceived social support was a predictor of prosocial behaviors, explaining 24.4% and 12% of the variance, respectively. Therefore, perceived social support would be a protective factor that would promote prosocial behaviors in both men and women; this finding indicates the importance of enhancing social support under conditions of home confinement, either physically with cohabitants or remotely through video calls, voice calls, or other forms of contact [5,7].

Regarding the unconfined groups, anecdotal evidence was found in favor of the nonexistence of this correlation, contrary to what was expected [5,7]. Despite this, the interpretation of the Bayes factors obtained does not allow us to state that this relationship does not exist, since only anecdotal evidence in favor of this possibility was found [32,34].

The second hypothesis proposed was that prosocial behaviors would be directly related to people’s capacity for resilience, with the latter being a protective factor that would enhance the former. It was hypothesized that this relationship would occur in both men and women, confined and unconfined. The results obtained support this hypothesis. In the case of women, extreme evidence was found for a direct correlation between both variables, and that resilience would act as a predictor of prosocial behaviors in both the confined and unconfined groups, explaining 7% and 8.8% of the variance in prosocial behaviors, respectively. In the case of men, the results showed moderate evidence of these relationships in the confined and anecdotal evidence of these relationships in the unconfined, with resilience explaining 8.4% and 5.1% of the variance in prosocial behaviors, respectively. The results support the existence of a direct relationship between prosocial behaviors and resilience [9,10], and also support the theory that resilience is a protective factor that favors this type of behavior [5,11].

## 5. Conclusions

This research has some limitations that are important to bear in mind when interpreting the results. First, as this is a cross-sectional study in which the data of confined and unconfined persons were collected independently, it is not possible to compare the scores obtained by the groups, or to affirm that confinement has increased or decreased any of the relationships analyzed. Furthermore, it is important to bear in mind that the exceptional situation of home confinement made it necessary for this part of the sample to be collected online, which may affect the results obtained.

Another limitation would be the low reliability of the BRCS. This instrument was chosen because it showed good psychometric properties in its validation in the Spanish population, but it would be interesting to carry out a study on the psychometric properties obtained in different samples.

Additionally, the specificity of the sample means that the results cannot be generalized to other population groups, and in future research, it would be interesting to add other variables such as the social origins of the participants or the conditions in which they passed the confinement.

In future studies, the option of carrying out a longitudinal study could be considered to resolve the limitations described above. With this type of research, it would be possible to compare the scores and relationships of the confined and nonconfined groups, and it would be important to take all measurements in person if health conditions permit, to guarantee the reliability of the results.

Finally, not having found a positive correlation between perceived social support and prosocial behavior in the unconfined group is an unexpected result. In future investigations, it would be necessary to verify if it is an isolated result of this sample, or on the contrary, if there is a difference in the relationship between both variables that depends on the conditions in which the people find themselves.

The results obtained in the present investigation support the theories that place social support (with the exception of the nonconfined group) and resilience as protective factors that favor prosocial behaviors. In addition, the most novel aspect of this research is that it provides data on the relationships among people who are living in home confinement, a condition that has not been investigated to date.

The practical implications of this research are focused on prosocial behaviors. Deepening the knowledge of these behaviors is important for psychology, since it has been proven that they are a key element in the proper functioning of societies, and in people’s health; even more so in crisis conditions such as the one we are experiencing worldwide with COVID-19. Although it is important to know the mechanisms involved in prosocial behaviors in order to improve the quality of life of people and the functioning of society, it is not less important to know if these processes are maintained under extreme conditions for society, such as living through a pandemic serious enough to decree home confinement for months. Knowing the functioning of constructs such as prosocial behaviors, resilience, or social support perceived by people during crises such as COVID-19 will help in the development of programs to guarantee the proper functioning of societies during future crises.

## Figures and Tables

**Table 1 healthcare-10-01669-t001:** Descriptive statistics.

Situation	Sex	Variable	N	Mean (SD)
No confinement	Woman	Social support	153	4.49 (0.54)
		Resilience	153	3.45 (0.70)
		Prosocial behavior	153	3.29 (0.43)
	Man	Social support	105	4.24 (0.69)
		Resilience	105	3.68 (0.63)
		Prosocial behavior	105	3.07 (0.52)
Confinement	Woman	Social support	228	4.46 (0.60)
		Resilience	228	3.61 (0.55)
		Prosocial behavior	228	4.20 (0.41)
	Man	Social support	84	4.12 (0.78)
		Resilience	84	3.75 (0.63)
		Prosocial behavior	84	3.85 (0.60)

**Table 2 healthcare-10-01669-t002:** Bayesian correlation matrix with prosocial behavior.

Situation	Sex	Variable	Pearson Correlation	Bayes Factor	Mode	Mean	Variance	CI
NC	Woman	SS	0.160	2.204	0.163	0.157	0.006	0.003, 0.309
		R	0.306	0.010	0.311	0.300	0.005	0.155, 0.440
	Man	SS	0.194	1.817	0.199	0.188	0.009	0.007, 0.369
		R	0.244	0.558	0.250	0.237	0.008	0.058, 0.413
C	Woman	SS	0.497	0.000	0.501	0.491	0.003	0.392, 0.588
		R	0.273	0.003	0.276	0.269	0.004	0.148, 0.387
	Man	SS	0.362	0.039	0.372	0.349	0.009	0.161, 0.530
		R	0.309	0.199	0.318	0.297	0.010	0.106, 0.488

The values in this table are presented to 3 decimal places for a better understanding of the results. The analysis assumes uninformative Jeffreys a priori distributions. The mode, mean, and variance are for the a posteriori distribution. NC—no confinement; C—confinement; SS—social support; R—resilience; CI—95% credibility interval (lower limit, upper limit).

**Table 3 healthcare-10-01669-t003:** Bayesian estimates of regression coefficients.

Situation	Sex	Relation	Mode	Mean	Variance	CI
NC	Woman	PB ← SS	0.128	0.128	0.004	0.001, 0.254
		PB ← R	0.186	0.186	0.002	0.093, 0.279
	Man	PB ← SS	0.146	0.146	0.005	0.002, 0.289
		PB ← R	0.203	0.203	0.006	0.046, 0.360
C	Woman	PB ← SS	0.337	0.337	0.002	0.260, 0.415
		PB ← R	0.202	0.202	0.002	0.108, 0.295
	Man	PB ← SS	0.279	0.279	0.006	0.121, 0.437
		PB ← R	0.298	0.298	0.011	0.096, 0.499

The values in this table are presented to 3 decimal places for a better understanding of the results. The analysis assumes uninformative a priori distributions, and the JZS method is used. The mode, mean, and variance are for the a posteriori distribution. NC—no confinement; C—confinement; PB—prosocial behavior; SS—social support; R—resilience; CI—95% credible interval (lower bound, upper bound).

## Data Availability

The raw data supporting the conclusions of this article will be made available by the authors, without undue reservation.

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
