# Peer review of "Social Support and Resilience as Predictors of Prosocial Behaviors before and during COVID-19"

_healthcare, 2022, doi:10.3390/healthcare10091669_

Round 1
Reviewer 1 Report
The authors aimed to analyze the relationship between social support and resilience with prosocial behavior before and during the confinement caused by COVID-19.
They recruited a sample of participants.
The participants were divided into a confined group (228 women and 84 men) and an unconfined group (153 women and 105 men), all of whom were university students. Instruments were applied to measure the variables proposed.
They found that : (a) social support predicted 24.4% of the variance in prosocial behavior among women and 12% among men in the confined group; no evidence of this relationship was found in the unconfined groups. (b) Resilience predicted 7% of the variance in prosocial behavior among confined women, 8.4% among confined men, 8.8% among unconfined women and 5.1% in unconfined men.
The authors concluded that the results show the importance of social support and resilience in prosocial behaviors, which are key elements for the proper functioning of society, especially in the face of a crisis such as COVID-19.
This is a well written article falling in a hot topic.
I have some minor comments for the authors:
1. The abstract must contain the summary of all the sections (for example the background)
2. This study has interesting perspectives. Please enlarge them in the discussion
3. Insert the conclusions
4. Perhaps a flow chart in the method could aid the readability of the scientific design
Author Response
July 25, 2022, Mexico City
Responses to reviewers
Social Support and Resilience as Predictors of Prosocial Behaviors before and during COVID-19
Prof. Dr. Paul B. Tchounwou
Editor-in-Chief
Thank you for the opportunity to resubmit our manuscript ID: healthcare-1816145 "Social Support and Resilience as Predictors of Prosocial Behaviors before and during COVID-19" to Healthcare. We appreciate the opportunity to publish in Healthcare. We have carefully reviewed the reviewers' valuable comments. To ensure that we have fully addressed all of your concerns, we have revised our manuscript based on the reviewers' suggestions.
Below, we include our point-by-point responses to the reviewers' corrections and comments and describe in detail the changes made to the manuscript.
Thank you again for the opportunity to resubmit our manuscript. I appreciate any help you can provide.
Sincerely,
Filiberto Toledano-Toledano, Ph.D.
Federico Gómez Children’s Hospital of Mexico, National Institute of Health.
Dr. Márquez 162, Doctores, Cuauhtémoc, México City, 06720, México.
+ 52 55 52289917, ext. 4318. E-mail: filiberto.toledano.phd@gmail.com
Reviewer 1
Comments and Suggestions for Authors
The abstract must contain the summary of all the sections (for example the background)
Response:
Thank you very much for all your comments to improve our manuscript. We are glad that you found it an interesting topic.
We have modified the abstract to include all section headers.
Change:
Headers have been included in the abstract
This study has interesting perspectives. Please enlarge them in the discussion
Response:
Heeding their advice, we have improved the conclusions of the manuscript to highlight the importance of the results.
Change:
“Although it is important to know the mechanisms involved in prosocial behaviors in order to improve the quality of life of people and the functioning of society, it is not less important to know if these processes are maintained in extreme conditions. for society, such as living through a pandemic serious enough to decree home confinement for months. Knowing the functioning of constructs such as prosocial behaviors, resilience or social support perceived by people during crises such as COVID-19, will help develop programs to guarantee the proper functioning of society during future crises.”
Insert the conclusions
Response:
We have divided the discussion into discussion and conclusions following your indications
Change:
The "conclusions" section has been included
Perhaps a flow chart in the method could aid the readability of the scientific design
Response:
We highly value your input and have tried to make changes to include the flow chart, but once included we felt that the information was somewhat redundant.
Change:
None.
Reviewer 2 Report
The proposed contribution addresses a highly topical societal issue by comparing the effects of social support and resilience on prosocial behaviors in an unconfined period (pre-Covid) and in a confined period. The problematization is well related to the literature and to the questions posed therein. The hypotheses are clearly stated and the methodology clearly presented, both from the point of view of the samples and of the questionnaire. Finally, the statistical analysis clearly demonstrate on the one hand that in the period of confinement, the perceived social support promotes prosocial behavior for both men and women and, on the other hand, that both in periods of confinement and of non-confinement, the capacity for resilience is a positive factor for prosocial behaviors among women. Finally, the authors underline two strong limits to their work: the fact that they were not able to follow the same cohort before and after confinement and the different conditions for administering the questionnaire in the two situations.
However, it is surprising that other points have not been raised. First, the fact that Cronbach's alpha values ​​are relatively low for the resilience scale (but this is a scale borrowed from other works). Above all, nothing is said about the specificity of the studiedpopulation – students – even though this choice strongly limits the possibility of generalizing the results. All the more so since the Covid period was experienced in a very specific – and very variable – way by the student population. Moreover, the study focuses on the correlations between the three main studied phenomena – while adding a gender variable – without taking into account other elements which could have a very strong influence on the results: social backgrounds, conditions of confinement (which cannot be reduce to percieved social support or not),… Finally, certain results which do not or do not really agree with the study are largely set aside, without questioning. For example, the negative correlation, in the study, between perceived social support and prosocial behavior outside the confined period. Even though the works generally show a positive correlation.
Without calling into question the interest of the study and its contributions, it would seem important to recall all of these limits, and even, sometimes, to provide answers to them.
Author Response
July 25, 2022, Mexico City
Responses to reviewers
Social Support and Resilience as Predictors of Prosocial Behaviors before and during COVID-19
Prof. Dr. Paul B. Tchounwou
Editor-in-Chief
Thank you for the opportunity to resubmit our manuscript ID: healthcare-1816145 "Social Support and Resilience as Predictors of Prosocial Behaviors before and during COVID-19" to Healthcare. We appreciate the opportunity to publish in Healthcare. We have carefully reviewed the reviewers' valuable comments. To ensure that we have fully addressed all of your concerns, we have revised our manuscript based on the reviewers' suggestions.
Below, we include our point-by-point responses to the reviewers' corrections and comments and describe in detail the changes made to the manuscript.
Thank you again for the opportunity to resubmit our manuscript. I appreciate any help you can provide.
Sincerely,
Filiberto Toledano-Toledano, Ph.D.
Federico Gómez Children’s Hospital of Mexico, National Institute of Health.
Dr. Márquez 162, Doctores, Cuauhtémoc, México City, 06720, México.
+ 52 55 52289917, ext. 4318. E-mail: filiberto.toledano.phd@gmail.com
Reviewer 2
Comments and Suggestions for Authors
However, it is surprising that other points have not been raised. First, the fact that Cronbach's alpha values ​​are relatively low for the resilience scale (but this is a scale borrowed from other works).
Response:
You're right, it's a major limitation for the research. In addition, it is an issue that surprised us and we are currently carrying out a reliability generalization meta-analysis to check if it is an isolated problem or if the instrument is not reliable.
Change:
“Another limitation would be the low reliability of the BRCS. This instrument was chosen because it showed good psychometric properties in its validation in the Spanish population, but it would be interesting to carry out a study on the psychometric properties obtained in different samples.”
Above all, nothing is said about the specificity of the studied population – students – even though this choice strongly limits the possibility of generalizing the results. All the more so since the Covid period was experienced in a very specific – and very variable – way by the student population. Moreover, the study focuses on the correlations between the three main studied phenomena – while adding a gender variable – without taking into account other elements which could have a very strong influence on the results: social backgrounds, conditions of confinement (which cannot be reduce to percieved social support or not). Finally, certain results which do not or do not really agree with the study are largely set aside, without questioning. For example, the negative correlation, in the study, between perceived social support and prosocial behavior outside the confined period. Even though the works generally show a positive correlation.
Response:
We have included these aspects among the limitations of the research.
Change:
“Additionally, the specificity of the sample means that the results cannot be generalized to other population groups and, in future research, it would be interesting to add other variables such as the social origin of the participants or the conditions in which they passed the confinement.”
“Finally, not having found a positive correlation between perceived social support and prosocial behavior in the unconfined group is an unexpected result. In future in-vestigations, it would be necessary to verify if it is an isolated result of this sample or, on the contrary, if there is a difference in the relationship between both variables de-pending on the conditions in which the people find themselves.”